



# On the impact of thunder on cloud ice crystals and droplets

Konstantinos Kourtidis[1,*], Stavros Stathopoulos[1], Vassilis Amiridis[2]

[1]Department of Environmental Engineering, Democritus University of Thrace, 671 00 Xanthi, Greece
[2]Institute for Astronomy, Astrophysics, Space Applications and Remote Sensing, National Observatory of Athens, 152 36 Athens, Greece

*Correspondence to*: Konstantinos Kourtidis (kourtidi@env.duth.gr)

**Abstract.** Calculations are presented on the impact of thunder on cloud particles. The results show that the creation of a shockwave front near the lightning channel results in shattering of ice crystals, droplets, and dust aerosols, the former being a
yet unidentified mechanism for secondary ice production in clouds. At low altitudes shattering is more efficient. At the distance where the shockwave front decays to audio wave, it can cause agglomeration of particles. The cloud particles' characteristics appear not very suitable for extensive acoustic agglomeration if the Sound Pressure Level (SPL) is below 120 dB. Nevertheless, even for SPL<120 dB, some agglomeration will occur. Agglomeration will occur readily if SPL>135 dB at sound frequencies 10-200 Hz. Agglomeration efficiency increases with height. More agglomeration will occur in pyroclouds, due to their large
particle number densities. These results show that the electrical environment in clouds has, through thunder, effects on the size distribution and number density of ice particles and droplets, will hence influence thundercloud radiative properties, and it may be a significant driver of secondary ice production. As global warming may influence the occurrence rate of lightning, the mechanisms discussed here may induce a climate feedback.

**1 Introduction**

Cloud droplet size distribution affects precipitation and the radiative effects of clouds. The effect of particulate matter on cloud droplet size distribution and precipitation rate has been investigated extensively in the last two decades (e.g. Stier et al., 2024). Ice clouds are ubiquitous in the global atmosphere, making up to 70% of clouds in the tropics. By absorbing longwave radiation and scattering shortwave radiation, they may warm or cool Earth's surface (Yang et al., 2015). The size distribution of ice
particles is a crucial parameter not only with regard to longwave absorption and shortwave scattering efficiency (Liu et al., 2014) but also for precipitation rate. Reducing the uncertainty in ice particle size can result in large improvements in modeling climate sensitivity to increasing $CO_2$ concentrations in climate models (Wang et al., 2020). Observed ice crystal concentrations often exceed the concentration of ice nucleating particles (INPs) by orders of magnitude. Secondary ice production (SIP) may be very important in controlling the ice crystal concentrations (e.g. Field et al., 2017; Korolev and Leisner, 2020).
In the present work, we examine the effect that thunder can have on SIP and on the size distribution of cloud droplets.
Sonic flows are in use for decades for the breakup of large droplets in many practical applications, such as combustion, gasification, emulsification and medicine (e.g. Jain et al., 2015; Zhao et al., 2019, and references therein). Hanson et al. (1963)



were among the first to investigate the breakup of droplets by air blasts. It is now well known that liquid droplets can undergo deformation and breakup if exposed to a gas stream of sufficient velocity (e.g. Wierzba, 1990; Guildenbecher et al., 2011). On the other hand, acoustic agglomeration of very small particles is also in practical use in air pollution control technologies. It has been studied for some decades now as an effective means for removing fine particles from industrial gaseous effluents (e.g. Scott, 1975; Hoffmann and Koopmann, 1996; Gallego-Juárez et al., 1999; Ng et al., 2017) or automotive exhaust (De Sarabia et al., 2003; Noorpoor et al., 2012) by coagulating them into coarser particles.

Very recently, interest has emerged, especially in China, for using artificial sounds to induce droplet coalescence in fog or clouds and hence disperse fog or induce rain. Liu et al. (2020) simulated numerically the condensation of fog under sound of 140-160 dB SPL and frequencies of 100-2000 Hz. Qiu et al. (2021) in a cloud chamber experiment with droplets of 10 μm diameter observed effective agglomeration of water droplets for Sound Pressure Levels (SPL) 114-121 dB in the 50-65 Hz range. Unfortunately, they do not report the number concentrations of droplets used in the experiment. Jia et al. (2021) provided a theoretical examination of the effect of strong sound waves on cloud droplets. Wei et al. (2021) used artificially generated sounds in the field to induce droplet agglomeration in clouds. Bai et al. (2022) contacted laboratory experiments and simulations on the action of sound waves on microdroplets, using SPL of 70-130 dB and frequencies 30-280 Hz, while Shi et al. (2022) conducted extensive field tests with 10 kW speakers and output levels of 148.6 dB in order to study artificial rain production.

Very few people, however, have examined the effect of ambient sounds on droplets and aerosols. Recently, Kourtidis and Andrikopoulou (2022) examined whether bell sounds can have an impact on the size distribution of ambient aerosol. Some time ago, Temkin (1969) was the first to suggest, in a half-page paper, that thunder may induce collisions in cloud droplets and hence simulate droplet growth. Recently, Temkin (2021) presented calculations for droplet coalescence induced by the thunder sound, using a theoretically obtained value of 8 Hz as the dominant thunder clap frequency, and found that droplet agglomeration occurs.

Lightning occurs in cloud environments where not only large numbers of droplets but also ice nuclei are present. Thunder may induce mechanical effects on atmospheric particles, as the SPL can be quite high, and, additionally, thunder frequency spectra have peaks at low frequencies, where orthokinetic agglomeration is known to be very effective (e.g. Dong et al., 2006). Another mechanism that may influence the size distribution and number density of cloud particles, and has not been studied until now, is the supersonic shockwave front that results from the rapid heating of air to several tens of thousands degrees. The supersonic shockwave operates for some distance from the lightning channel, after which, the shockwave turns to sound wave.

In the present work we examine the effects of thunder on the size distribution of cloud droplets and ice nuclei. We will investigate not only droplet coalescence but also droplet and ice nuclei breakup in the thunder shockwave front. This is the first time the latter is studied.

## 2 Results and discussion

### 2.1 Particle breakup in the supersonic thunder shockwave front



Up to now, only Goyer et al. (1965) has studied effects of the thunder shockwave, investigating in the laboratory possible shock-induced freezing of super-cooled water droplets. Here, we will examine a different effect of the thunder shockwave,

namely the possibility for particle breakup by the rapid expansion of air in the lightning channel. During a lightning discharge, deposition of energy in the 4-100 J/cm range (Stark et al., 1996; Borovsky, 1998; Lacroix et al., 2019), heats within a few μs air to $10^5$ K plasma, resulting in very rapid expansion of air.

The non-dimensional Weber number is defined as $We=\rho_g \upsilon^2 d/\sigma$ , where $\rho_g$ is the air density in kg m$^{-3}$, $\upsilon$ is the relative air velocity in m s$^{-1}$ between gas and particle, d is the particle diameter in m and σ is the surface tension (surface energy, for solids) in N

m$^{-1}$. Experiments show that droplets placed suddenly in a high speed air flow will break up if the Weber number exceeds ~12 (e.g. Krzeczkowski, 1980; Wierzba, 1990; Zilch et al., 2008). This number is the critical Weber number, $We_{cr}$. The modified Weber number We*=We/12, equals the ratio of the kinetic energy on impact to the surface tension (surface energy, for solid particles). Hence, when the Weber number exceeds $We_{cr}$ , the kinetic energy on impact is higher than the surface tension (or energy).  We will use below the Weber number to examine if the conditions in the thunder shockwave lead to breakup of cloud

particles (droplets, ice crystals, aerosols). To this end, the relative air velocity, which equals the shockwave front velocity, is a crucial parameter. So, we will first review the sparse literature on the matter, to obtain as realistic as possible front velocities for the calculations.

Navarro-González et al. (2001) simulated lightning in the laboratory by generating hot plasma with a pulsed Nd-YAG laser, and determined shockwave front velocity of about 60 km s$^{-1}$ at 20 ns after the laser pulse. After about 3 μs, they observed

decoupling of the resulting supersonic shockwave from the plasma, and the shock front cooled off to near ambient temperatures at around 5 μs, where it propagated at near sonic speed. In agreement with these results, Stark et al. (1996) also simulated lightning in the lab and found shockwave velocities of 2.2 km s$^{-1}$ at 1 μs after the discharge and decreasing front velocity to below 1 km s$^{-1}$ after a few μs. Since the database on the shockwaves of lightning discharges is rather limited, we quote here also some relevant results from explosions. Jenkins et al. (2013) derived experimentally, using a high speed framing camera

and particle image velocimetry, velocities of particles after explosions of 1.3-1.7 km s$^{-1}$. Lacroix et al. (2019) theoretically derived particle velocities after explosions of ~0.3 km s$^{-1}$, while model results by Karch et al. (2018) show velocities 5 to >10 times the speed of sound, i.e. 1.7 km s$^{-1}$ to >3.4 km s$^{-1}$.

Liu and Chang (2014), by generating spark discharges in the laboratory, found a linear relationship between electric discharge energy and shockwave energy. Since the shockwave energy will be related to its expansion speed, given the wide range of

discharge energies observed in natural lightning, we will use in the calculations below shockwave front velocities of 60 km s$^{-1}$ and 1 km s$^{-1}$, which covers the range of observed front velocities, and will cover also a range of distances from the lightning channel.

For a 10μm cloud droplet, and a shockwave front velocity of 60 km s$^{-1}$ in the immediate vicinity of the lightning channel, *We* equals several hundreds of thousands. For this particle size, even when the front velocity drops to 1 km s$^{-1}$, for a droplet

We=167, for an ice particle We=63, and for a solid Al$_2$O$_3$ particle We=71.



In Table 1, calculations are presented for the minimum size of particles for which $We=12$ and $We=120$ will be exceeded. For Weber numbers near 12, the particles resulting from breakup have a bimodal distribution with a primary peak at $d/d_0=0.03$ (where d is the diameter of the particles resulting from breakup of particle with initial diameter $d_0$) and a secondary peak at $d/d_0=0.06$ and a Sauter mean diameter/$d_0$ around 0.09, whereas for Weber numbers near 120, the particles resulting from

breakup have a unimodal lognormal distribution with a peak at $d/d_0=0.023$ and a Sauter mean diameter/$d_0$ around 0.03 (Jain et al., 2015). We note here that the Sauter mean diameter (Sauter, 1926) is the diameter of a drop having the same volume to surface area as the entire spray.

Calculations are for the following types of particles: cloud droplets (for which the surface tension of pure water, 0.072 N m$^{-1}$, is used), ice crystals (for which the surface energy of pure water ice, 0.19 N m$^{-1}$, is used, after Gundlach et al., 2011), solid

$Fe_2O_3$ ($\sigma = 1.357$ N m$^{-1}$) and $Al_2O_3$ particles ($\sigma = 0.169$ N m$^{-1}$), and $SiO_2$-methanol particles (surface tension 0.023 N m$^{-1}$ from Bhuiyan et al., 2015). Solid $Fe_2O_3$ (iron(III) oxide, hematite) and $Al_2O_3$ particles are used as a proxy for dust, e.g. Saharan dust. $SiO_2$-methanol is used as a proxy for a dust particle covered with secondary organic aerosol (SOA). Above the minimum particle size depicted in Table 1, $We$ will exceed 12 (or 120). The calculations for ground level and 5 km ASL differ by the density of air, whereas possible changes in the surface energy of the particles due to the lower temperatures at 5 km ASL, have

not been taken into account, since they are generally small. Since surface tension increases slightly with decreasing temperature, the results for 5 km ASL presented in Table 1 underestimate the radii very slightly. Calculations are for front velocities range 60 km s$^{-1}$ to 1 km s$^{-1}$.

**Table 1. Minimum particle diameters (nm) in the thunder shockwave front (for front velocities of 60 km s$^{-1}$ and 1 km**

**s$^{-1}$) where the Weber number (We) becomes equal to the critical Weber number We$_{cr}$ = 12, or We=120.**

| | Ground level 60km/s – 1km/s | 5 km ASL 60km/s – 1km/s | $We$ |
|---|---|---|---|
| **cloud droplet** | 0.20 – 720 | 0.40 – 1.44·10³ | 12 |
| **ice crystal** | 0.54 – 1.95·10³ | 1.03 – 3.71·10³ | 12 |
| **solid Al$_2$O$_3$ particle ($\sigma = 0.169$ N m$^{-1}$)** | 0.47 – 1.7·10³ | 0.94 – 3.38·10³ | 12 |
| **solid Fe$_2$O$_3$ particle ($\sigma = 1.357$ N m$^{-1}$)** | 3.77 – 13.6·10³ | 7.54 – 27.14·10³ | 12 |
| **SiO$_2$-methanol particle** | 0.06 – 230 | 0.12 – 460 | 12 |
| **cloud droplet** | 2 – 7.2·10³ | 4 – 14.4·10³ | 120 |
| **ice crystal** | 5.43 – 19.55·10³ | 10.32 – 37.14·10³ | 120 |
| **solid Al$_2$O$_3$ particle ($\sigma = 0.169$ N m$^{-1}$)** | 4.7 – 16.9·10³ | 9.4 – 33.84·10³ | 120 |
| **solid Fe$_2$O$_3$ particle ($\sigma = 1.357$ N m$^{-1}$)** | 37.7 – 135.7·10³ | 75.4 – 271.44·10³ | 120 |
| **SiO$_2$-methanol particle** | 0.6 – 2.3·10³ | 1.2 – 4.6·10³ | 120 |



We note that most SOA particles have a lower surface tension that water, hence SOA will break up more easily than pure water droplets. Ice crystals and $Al_2O_3$ particles must have about 3 times the diameter of a water droplet to break, while $Fe_2O_3$ particles are the most difficult to break. At ground level, sub-nanometer cloud particles, and nanometer-sized $Fe_2O_3$ dust aerosols $We$

will reach the critical number 12, and hence they will break up in thunder shockwave fronts expanding at 60 km s$^{-1}$. Even at fronts expanding at 1 km s$^{-1}$, sub-micron cloud droplets and $SiO_2$-methanol particles, and small μm-sized ice crystals and $Al_2O_3$ particles will break, while $Fe_2O_3$ particles must be larger than 13.6 μm to break. In thunder shockwave fronts expanding at 60 km s$^{-1}$, $We$=120 will be reached for nanometer sized particles. At $We$=120, catastrophic breakup will occur and more secondary particles will be generated than for $We$=12. For a lower shockwave front velocity of 1 km s$^{-1}$, $We$=120 will be reached for all

types of particles larger than around 20 μm, except $Fe_2O_3$ particles that need to be larger than 136 μm to undergo catastrophic breakup.

At 5 km ASL, particles double the size of the ones at ground level will break. So, for lightning channels extending vertically, near the lightning channel a vertical gradient in the size distribution of cloud particles will be introduced.

With the extreme scarcity of data on the possible extend of the shockwave, it is not possible to evaluate how large are the parts

of the cloud that are affected from the shockwave. Goyer and Plooster (1968) using a numerical model of lightning discharge, calculated shock waves in the order of a few meters. Karch et al. (2018) simulated a 96.4 kA strike (i.e. 0.76 X 10$^4$ J m$^{-1}$) and found the shock wave transitioning to acoustic velocities at around 6 cm. Takagi et al. (1998) observed return lightning strokes with a high-speed camera and found that the luminous region expands at about 100 km s$^{-1}$ during the initial stage and reaches a maximum diameter of several meters after about 100 μs.

If the Karch et al. (2018) 6 cm shockwave radius is used, then assuming a cylindrical geometry it is easy to calculate that the shockwave from a 500 m long intra-cloud (IC) discharge will affect a volume of 5.65 m$^3$ within the cloud. Although this volume is small, multiple IC lightning discharges are common within thunderclouds and will increase it substantially. If we use the Goyer and Plooster (1968) calculations, or if the shockwave extends at the same distance as the luminous region, we can assume that the shockwave extends ~3 m from the channel. Then a 500 m long IC discharge will affect a volume of the

order of 14 X 10$^3$ m$^3$ within the cloud, which is substantial. Given also the fact that the total acoustical power of thunder was estimated by Bestard et al. (2023), from the study of 78 flashes, to span four orders of magnitude from 10.6 kW to 165 MW, both of the above calculations appear credible.

Given the importance of the ice phase in clouds in precipitation over the continents (Heymsfield et al., 2020), the importance of secondary ice production (SIP) in the formation of ice particles (Korolev and Leisner, 2020), and the need to properly

describe SIP in climate and weather models, the above mechanism may need to be taken into account in numerical descriptions.

## 2.2 Particle agglomeration in the thunder sonic field



After the supersonic wave front loses velocity, at some point, it turns to an expanding sonic field. There have been several
subjective terms such as clap, peal, roll and rumble to describe thunder sounds. Peals or claps are the sudden loud sounds
which occur in a background of prolonged roll or rumble. The term roll is sometimes used to describe irregular sound variations
whereas rumble is used to describe relatively weak sound of long duration (Depasse, 1994). Finite amplitude propagation
causes a doubling in the wavelength of the positive pulse within the first kilometer, but beyond this range, the wavelength
remains approximately constant (Few, 1995). As the SPL can be very high, thunder may induce mechanical effects on
atmospheric particles (Few et al., 1967).

Sound is known to cause agglomeration at high SPL, termed acoustic agglomeration or acoustic coagulation, due to particle
resonance and the resulting relative motion of particles. So, loud sound can impact atmospheric particles. We will examine
here acoustic agglomeration due to the sound of thunder. The main identified mechanisms for agglomeration are orthokinetic
collision and hydrodynamic collision. Orthokinetic collision is the main mechanism of sonic agglomeration for polydisperse
particles at low sound frequencies and medium particle size ratios $d_1/d_2$. The orthokinetic mechanism refers to collisions
between differently sized particles located within a distance that is approximately equal to the displacement amplitude of the
acoustic field and with their relative motion parallel to the direction of vibration (Riera et al., 2015). It is based on the different
resonance rate $\eta$ of the particles due to their different sizes, different displacement amplitudes for different sizes resulting in
increased collisions). The resonance of particles in a sonic field can be characterized by the resonance rate

$$\eta = U_p/U_o = 1/[sqr(1+(\omega\tau_p)^2)] \quad (1)$$

(Temkin and Leung, 1976; Hoffmann and Koopmann, 1996, 1997; Gonzalez et al., 2000), where $\eta$ the resonance rate with
values from 0 (no resonance) to 1 (complete resonance), $\omega$ is the sound wave angular velocity, $\tau_p$ is the relaxation time
$\tau_p = \rho * d^2/0.00032886$, $\rho$ is the particle density, $d$ is the aerodynamic diameter of the particle. The sound wave angular velocity
is given by $\omega = 2*\pi*f$, with $f$ being the sound frequency. Equation (1) is the simplified Brandt–Freund–Hiedemann (BFH)
equation (Brandt et al., 1936; González et al., 2000; Dong et al., 2006). The maximum interparticle distance that in the event
of a collision leads to agglomeration is the effective agglomeration length, $L_{eff}$. $L_{eff} = \varepsilon \cdot L$, where $\varepsilon$ is the collision efficiency
(with values between 0 and 1) and $L$ is the maximum interparticle distance that can cause collision. The value of $\varepsilon$ is controlled
by the Stokes number $S_t$, $\varepsilon = [S_t/(S_t+A)]B$ where A and B are constants (Löffler, 1988).

For the simulations presented here, we assumed liquid droplets with diameters d = 8-36 µm (e.g. Barthlott et al., 2022 and
references therein), and density $\rho$ = 1000 kg m$^{-3}$. Saharan dust particles have diameters 0.01-20 µm, with mass peaks at around
0.4 and 3 µm (Gini et al., 2022), number concentration peaks at around 0.03 µm, and surface area peak at around 10 µm
(Weinzierl et al., 2006). So, in the calculations presented here, for Saharan dust we used diameters d in the 0.1 – 10 µm range,
and density $\rho$ = 2500 kg m$^{-3}$, which is very near the density of silica.

Few et al. (1967) determined dominant thunder frequencies in the 180-260 Hz range. Holmes et al. (1971), by analyzing 40
thunder events, determined peak power at frequencies from below 4 Hz to 125 Hz. Intra-cloud (IC) discharges had a mean
peak value of power at 28 Hz with mean total acoustic energy 1.9·10$^6$ J, while cloud-to-ground (CG) discharges had a mean
peak value of power at 50 Hz with a much higher mean total acoustic energy of 6.3·10$^6$ J. Juhua and Ping (2012) observed





peak frequencies 210-280 Hz, while the frequency spectrum went up to 1000 Hz. The same authors, also calculated that the more powerful the lightning, the lower its peak frequency. Abegunawardana et al. (2016) determined fundamental frequency

of peals at 75±22 Hz, of claps at 102±36 Hz, and of rumbles at 63±27 Hz. Bodhika et al., 2014 observed thunder frequency spectra with peaks for peal and claps at around 100 Hz, and around 50 Hz for thunder rumble sounds. Lacroix et al. (2018) reported spectra in the 1-200 Hz range, which show amplitudes of 88-90 dB in the 6-80 Hz range, for flash distance 2-4.3 km. The same authors also report spectra from 14 events, which exhibit a frequency center of gravity that spans from 47 to 115 Hz, for different events.

Hence, for the calculations presented here, we use frequencies of 10-500 Hz.

In general, the resonance rate increases with decreasing frequency and decreasing particle size (Fig. 1). The results show complete or almost complete (>0.8) resonance for cloud droplet diameters =< 24 μm for f =< 70 Hz, while smallest 8 μm droplets remain completely resonant up to 200 Hz. Saharan dust particles with diameters up to 3 μm show complete resonance for f up to 500 Hz, while all dust particles up to 10 μm show complete or almost complete (>0.8) resonance for frequencies of

100 Hz or lower.

To calculate $L_{eff}$, the SPL is needed. Not very many SPL measurements exist in the vicinity of thunder. Bodhika et al. (2018) observed peak SPL above 110 dB for 30% of recorded flashes at 3 km from the flash. Closer to the lightning higher SPL values are to be expected. If only the geometrical spreading of a spherical wave in free space is considered, at half the distance the SPL would be increased by 6 dB. In a real atmosphere, the sound propagation deviates from spherical shape, and ground

reflection may increase the SPL, hence the difference may be less than 6 dB. Farges et al. (2021) found a decay of the thunder amplitude to scale with flash distance as $r^{-0.717}$, while Shi et al. (2022) report SPL(dB) decay rates scaling with distance at $r^{-0.06339}$. Lacroix et al. (2019) calculated overpressures 2 m from the stroke that translate to SPL 134-151 dB (for deposited energy 4-60 J/cm). Lacroix et al. (2019) also note a near- and a far-field behavior, acoustic power scaling with $r^{-1}$ with distance (cylindrical wave decay) up to 3600 m and scaling with $r^{-2}$ (spherical wave decay) after that. So, it is reasonable to assume that

the thunder SPL will exceed 120 dB at distances less than 800 m from the stroke and 130 dB at distances less than 200 m from the stroke.  Hence, for the calculations presented hereafter, we will use SPL in the 90-135 dB range.







**Figure 1: Upper panel: Resonance rate *η* of cloud droplets with diameters 8-36 µm (calculation step 2 µm) for sound frequencies 10-500 Hz (calculation step 10 Hz up to 100 Hz and 100 Hz above that). Lower panel: Resonance rate *η* of dust particles with diameters 0.1-10 µm (calculation step 1 µm) for sound frequencies 10-500 Hz (calculation step 10 Hz up to 100 Hz and 100 Hz above that).**





**Figure 2: Effective agglomeration length near the surface for SPL=100 dB and sound frequencies of 10 (top), 50 (middle), and 200 Hz (bottom).**






For two particles No. 1 and No. 2 (No. 1 being the larger particle and No. 2 the smaller one) with diameters $d_1$ and $d_2$, their relative resonance rate is given by $\eta_{12}=\omega*(\tau_{p1}-\tau_{p2})/\mathrm{sqr}[(1+\omega^2*\tau_{p1}^2)*(1+\omega^2*\tau_{p2}^2)]$, where $\tau_{p1}=\rho_1*d_1^2/0.00032886$ and $\tau_{p2}=\rho_2*d_2^2/0.00032886$ are the relaxation times of particle 1 and 2 and $\omega=2*\pi*f$ the angular velocity of the sound wave. The effective agglomeration length for these two particles is calculated by $L_{eff}= (\eta_{12}*U_g/\omega)*[S_t/(S_t+0.65)]^{3.7}$, where $S_t$ is the Stokes number $S_t=\rho_2*\eta_{12}*U_g *d_2^2/(0.00032886*d_1)$ and $U_g=\{10^{[(SPL-94)/20]}\}/(c*\rho_g)$, $U_g$ being the gas velocity amplitude in the sound wave, c the velocity of sound in air, and $\rho_g$ the density of air.

For two particles with $d_1$ in the 0.05-35 μm range and $d_2$ in the 0.02-30 μm range, for SPL=100 dB and frequencies of 10, 50 and 200 Hz, the effective agglomeration length spans many orders of magnitude, and is larger for larger particle pairs (Fig. 2). For particles above around 25 μm, it ranges between 1 nm and 10 μm (Fig. 2). Increasing the SPL from 90 to 120 dB, increases the agglomeration length by several (6-7) orders of magnitude (Figs. 3 and 4). Further increasing the SPL from 120 dB to 135 dB, increases further the agglomeration length by 3-4 orders of magnitude. Hence, the agglomeration length ~100 m from the strike will be 4 orders of magnitude longer than the agglomeration length ~800 m thereof and 6-7 orders of magnitude longer than the agglomeration length ~2 km thereof. Additionally, the agglomeration length is 10-50 times larger at 5 km ASL than near ground level (Fig. 4), hence in clouds with vertical extend like cumulus congestus and cumulonimbus, the higher levels of the cloud will experience more coagulation than the lower ones.

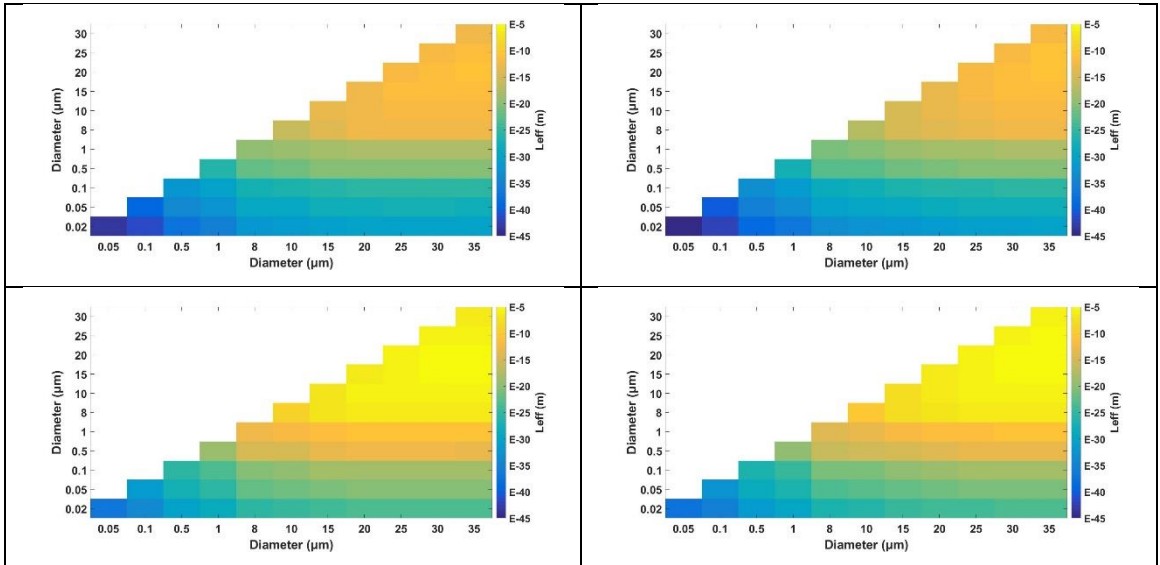

**Figure 3: Effective agglomeration length for pairs of particles with diameters 20 nm-35 μm near ground level for sounds of 100 Hz (left panels) and 50 Hz (right panels), and SPL 90 dB (top) and 120 dB (bottom).**

For number concentration of particles per unit volume N, the mean interparticle distance $<r>$ is proportional to the per particle volume 1/N. It can be defined as $<r>=1/N^{1/3}$, $<r>$ corresponding to the length of the edge of a cube of volume 1/N. For Saharan dust particles, for which $N_{dust} \sim$ 1-100 particles/cm$^3$, $<r>= \sim$ 2 - 10 mm. For cloud droplets $N_{cloud\_droplets}\sim$ 200-1000 droplets/cm$^3$,





hence $<r>= \sim 1$-1.7 mm, for ice particles $N_{ice\_particles} \sim 0.1$-50 particles/cm$^3$ and $<r>= \sim 2.7$mm – 2.15 cm. For pyroclouds, $N_{soot}$ $\sim 200$-$10^5$ particles/cm$^3$, $<r>= \sim 210$ μm-1.7 mm.



**Figure 4: Left panels: Effective agglomeration length for large particles of 0.1-35 μm diameters and small particles for 0.05, 1 and 10 μm diameter, both at ground level and at 5 km height ASL, for *f*=50 Hz and SPL of 90 dB (a), 120 dB (b) and 135 dB (c). Right panels: Effective agglomeration length for large particles of 0.1-35 μm diameters and small particles for 0.05, 1 and 10 μm diameter, both at ground level and at 5 km height ASL, for *f*=10 Hz and SPL of 90 dB (d), 120 dB (e) and 135 dB (f).**

The effective agglomeration length, can be up to 10 μm for SPL 120 dB (Fig. 4), hence interparticle distances appear much larger than $L_{eff}$ for agglomeration to occur for SPL up to 120 db. However, the use of mean interparticle distance as $1/N^{1/3}$ ignores the fact that interparticle distances will follow a probability density function $P(r)=(3/a)*(r/a)^2*e^{[-(r/a)^3]}$ and the



substantial in-cloud turbulence of thunderclouds will further enhance the spread of *P(r)*. So, a substantial number of cloud particles will be at interparticle distances <<< $1/N^{1/3}$ and thus even for SPL below 120 dB some agglomeration will occur.

For SPL 135 dB, the agglomeration length can be up to 10 μm for particles with d>15 μm, and for very large particles (d=35μm) it can reach 1 mm. So, cloud droplets will agglomerate readily, and even more so in pyroclouds.

Ice, due to its higher than liquid water surface energy, will aggregate upon impact more easily than liquid water droplets (Gundlach and Blum, 2015). However, the aggregation process will be more complicated for ice collisions than for cloud droplets, since the event of ice-ice collision may cause apart from agglomeration, also rime splintering, hence the formation of a larger particle may be accompanied with the ejection of many small splinters. The event of ice-liquid droplet collision, which is more likely due to the higher abundance of cloud droplets, may lead to the formation of a larger ice crystal by freezing of the droplet water.

Qiu et al. (2021) observed effective agglomeration for SPL 114-121 dB in the 50-65 Hz range in cloud chamber experiments with droplets with diameters 4-20 μm; unfortunately, they do not report the number concentrations of droplets used in the experiment. Bai et al. (2022) determined critical SPL 110 ± 15 dB for effective agglomeration of microdroplets in the 1-30 μm range. Bai et al. (2023) identified in laboratory experiments optimal orthokinetic agglomeration frequencies 50-250 Hz for microdroplets. An optimal frequency, varying for different droplets, was identified in orthokinetic agglomeration within the 50–250 Hz range from Shi et al. (2023). Our results are consistent with these studies. However, we note here that in contrast to the present study, these studies were not concerned with naturally occurring sounds such as thunder, but with man-made sounds deliberately produced for rain enhancement or fog dissipation.

Prior to the present study only Temkin (2021; 2023) presented calculations for the impact of thunder on cloud droplets. Temkin (2021), using 8 Hz as the dominant thunder clap frequency, calculated that droplet agglomeration will occur rapidly, while Temkin (2023) studied the combined effects of thunderclaps and gravity on rain production.

Reliable thunder SPL and frequency spectra measurements at very short distances (10 m – 1 km) from thunder will contribute to the refinement of the calculations, as will also studies that can help determine the shockwave extend. Field measurements within thunderclouds of cloud droplet and ice crystal concentrations before and after lightning will show the extent of the effect thunder has on the size distributions of cloud particles.

**3 Conclusions**

We present results that show that the occurrence of thunder has the potential to alter the number concentration and size distribution of ice particles and cloud droplets within thunderclouds. As global warming may change the occurrence rate of lightning (e.g. Romps et al., 2014; Clark et al., 2017; Finney et al., 2018), the mechanism discussed here may introduce a climate feedback.

The two mechanisms examined here have different impact. The first, operating at the shockwave front in the vicinity of the lightning channel, results in extensive shattering of cloud particles, so it increases the number of particles and decreases their



size. The mechanism may be important as a secondary ice production mechanism. The second mechanism, operating at larger
distances from the lightning channel, causes coagulation, so it decreases the number of particles and increases their size.

290 The first mechanism operates at the shockwave front of the lightning and will cause extensive break-up of cloud particles
larger than a given diameter, depending on their composition. SOA particles will break up more easily than pure water droplets.
Ice crystals and $Al_2O_3$ particles must have about 3 times the diameter of a water droplet to break, while $Fe_2O_3$ particles are the
most difficult to break. At ground level, sub-nanometer cloud particles and ice crystals, and nanometer-sized $Fe_2O_3$ dust
aerosols will break up in thunder shockwave fronts expanding at 60 km s$^{-1}$. Even at fronts expanding at 1 km s$^{-1}$, sub-micron
295 cloud droplets and $SiO_2$-methanol particles, ice crystals and $Al_2O_3$ particles larger than ~2 μm will break, while $Fe_2O_3$ particles
must be larger than 13.6 μm to break. At 5 km ASL, particles double the size of the ones at ground level will break. So, for
lightning occurring at various cloud heights, a vertical gradient in the size distribution of cloud particles will be introduced.
Data on the possible extend of the shockwave are extremely scarse, and give ranges from a few cm to a few meters, so it is not
possible to evaluate how large are the parts of the cloud that will be affected from the shockwave. Another uncertainty arises
300 from the very limited data on the speed of the expansion of the shockwave front, which give ranges from 1 km s$^{-1}$ to 100 km
s$^{-1}$ and result in uncertainties of 3 orders of magnitude as to the smallest size of the particles above which shattering occurs.
However, even at the lower end of the expansion speed, all types of particles, except solid $Fe_2O_3$ ones, will break-up if they
are larger than ~0,23-2 μm. As this is the first time this mechanism is investigated, there are no previous results to compare to.
The second mechanism operates at larger distances from the lightning channel and results in acoustic agglomeration of cloud
305 particles. Larger particles will agglomerate more readily than smaller ones, for SPL above 120 dB and sound frequencies 50-
200 Hz. This mechanism's efficiency increases with height by about a factor of 10-50 every 5 km. Reliable thunder SPL and
frequency spectra measurements at very short distances (10 m – 1 km) from lightning will contribute to the refinement of the
calculations. The results presented here compare well with the emerging body of evidence from laboratory and field studies
with artificial sounds. They are consistent with the only two previous studies investigating the coagulation impact of thunder
(Temkin 2021, 2023), but extend substantially both the frequency range (as Temkin investigated frequencies of 5 Hz and 8
Hz) and the size of particles (above around 20 μm for Temkin).
The two mechanisms described above are operating in tandem, and will cause also vertical changes in the size distribution of
cloud particles, as they have different efficiencies at different heights. The results presented here demonstrate that thunder has
the potential to alter the size distribution of cloud droplets and ice crystals in thunderclouds, and may be important in generating
secondary ice particles. As the size distributions of droplets and ice crystals influence the rain generation process on the one
hand and the radiative properties of clouds on the other, the thunder impact is worth investigating further. The results are also
relevant from an atmospheric electricity point of view. As the charge separation within thunderclouds is influenced by the size
distribution of cloud particles and the collisions between them, it is interesting that this charge separation, causing lightning,
also causes thunder that results then in collision enhancement and changes in the size distribution and hence might introduce
a yet unquantified feedback in the electrification process. The extent of the changes these mechanisms introduce can be further



quantified only with carefully designed field measurements. Field measurements within thunderclouds of cloud droplet and ice crystal concentrations before and after lightning, together with lightning location data and SLP measurements may show the real extent of the effect thunder has on the size distributions of cloud particles.

**Author contribution:** KK conceptualization, investigation, supervision, original draft preparation. KK, SS and VA formal analysis. SS visualization. SS and VA review and editing.

**Competing interests:** The authors declare that they have no conflict of interest.

**Acknowledgements:** Vassilis Amiridis (VA) has received funding from Horizon Europe programme under Grant Agreement No 101137680 via project CERTAINTY (Cloud-aERosol inTeractions & their impActs IN The earth sYstem).

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
