# Peer review of "On the impact of thunder on cloud ice crystals and droplets"

_EGUsphere, 2024_

## Referee Comment (RC2)

Comments on 'On the Impact of thunder on cloud ice crystals and droplets' by K. Kourtidis, S. Stathopoulos and V. Amiridis.

Summary

I have read with interest the manuscript. Although a bit long, it is a reminder of the possible effects that thunder may have on ice particles and droplets. However, for the reasons mentioned below I cannot recommend it for publication.

Main points (to be explained later)

Contrary to the article's title and conclusions, the article does not consider, by far, the impact of thunder on cloud particles. At best it is a parametric study of two of the effects they consider: Breakup in shock waves and agglomeration in an acoustic wave. But there are problems with both. The first is considered on the basis of an erroneous assessment of the air velocity in a shock; the second is parametrized on the basis of models and calculations that are not applicable to either clouds or thunder. Incidentally, the case they consider -sinusoidal acoustic wav, has been thoroughly studied in articles not cited by the authors.

Also important is the fact that the authors include, without explanation, certain results upon which they base their estimates. The main examples of this are two equations, both central to their arguments, which appear in Sec. 2.2. Because of that omission I include in my comments an extended discussion of the origin and limits of those equations.

Sec. 2.1 Particle breakup in the supersonic thunder shockwave front

The main point considered in this section is the breakup of droplets produced by shock waves. This effect is well-known and well-understood. The contribution of this section is Table 1, which shows their estimates for the minimum sizes of particles that will break up, using as a criterion the experimentally-obtained value for the Weber number that produces droplet breakup, $We_{cr}$, which they say is 12. This value is in the range of values that have been found (although the actual value depends on the type of breakup). The problem I have in this regard is that they take that value to apply to the breakup of solid particles. I know of no experimental research supporting that application.

In any event, the authors tabulate the minimum sizes of some types of particles that would breakup when exposed to shock waves whose fronts move at 60 km/s or 1 km/s. To find those sizes they use the definition of the Weber number for breakup, solving for the diameter, i.e,

$$d = \sigma We_{cr} / \rho_g u^2$$

Here $u$ is the air velocity. This is fine, but in getting the values of $u$ corresponding to those two shock waves, the authors make a fundamental error: According to them (line 80) 'the relative air velocity equals the front velocity.' This is incorrect. The front of a shockwave always moves faster than the gas behind it. This is particularly important for the shock speeds they have in mind. The issue here is not the numerical difference but the erroneous conception of what a shock wave is.

As for the shock speeds they use, 60 km/s and 1 km/s, I assume each may occur within a short distance from the lightning discharge, but I wonder about the effects on nm size ice particles of the high temperatures that exist behind the corresponding shocks, as well as on the molecular composition of air in those conditions. [Incidentally, a very good source of information about such matters is Zeldovich and Raizer, *Physics of Shock Waves and High Temperature Hydrodynamic Phenomena*, 2 Vols., Academic 1966.]

To conclude, there is nothing new in this section other than an erroneous estimate of the minimum droplet sizes that result in breakup and the questionable use of information obtained with water droplets for the case of solid particles.

Sec. 2.2 Particle agglomeration in the thunder sonic field

In this section the issue of agglomeration in 'the thunder sonic field' is parametrized in the basis of a procedure, called orthokinetic agglomeration by some authors. Although there are several variations, the technique refers to agglomeration of particles due to the relative velocities of small particles of different sizes responding to single-frequency sound wave . The literature is vast but a basic source is Mednikov, *Acoustic coagulation and precipitation of Aerosols*, Consultant Bureau, 1965. Most of the work in this area has been done with high-frequency sound waves, but some time ago Temkin (Droplet agglomeration in a shock wave flow field, *Phys. Fluids*, June1570, not cited) showed that agglomeration can be effectively produced with low frequency waves that have shock fronts.

As remarked later, much closer to the work in review is Temkin's study of agglomeration of polydisperse distributions by monochromatic sound waves (Temkin, Gasdynamic agglomeration of aerosols. I, Acoustic waves, *Phys. Fluids*, July 1994, not cited).

In this work, the authors use the term 'resonance ratio' for the ratio of the displacement (or velocity) of a particle in a sound field to the gas velocity in the wave, usually called the 'Entrainment Ratio' in the literature. In any event, the ratio as given by eq. 1, or

$$\eta = {U_p}/{U_0} = \left[ 1 + \left( \omega \tau_p \right)^2 \right]^{-1/2} \qquad (1)$$

Note that the two velocities appearing in this equation are not defined, nor is it stated anywhere in the text that the equation applies only in very limited cases: spherical particles, single-frequency sound waves, very small gas velocities (small Reynolds number) and frequencies that are not large.

The authors define the quantity $\tau_p$ appearing in their Eq. 1 is defined as

$$\tau_p = {\rho_p d^2}/{0.0003286}$$

This 0.00032886 is not only unpleasant, but it also obscures the fact that it includes the value of the (dynamic) viscosity coefficient of the gas ($\bar{\mu}$) sustaining the motion, with which that the equation should be written differently, for example, with $18\bar{\mu}$ instead of that number.

This $\tau_p$, called the relaxation time in the literature, is a simple time scale for small-amplitude motions. In any event, the issue is not the way $\tau_p$ is written or why is it called that way, but whether Eq. 1 is at least approximately valid for a parametric study of effects that supposedly take place in a cloud as a result of thunder.

In any case, Eq. 1 is apparently used in the article to obtain the basic relation that the authors use to estimate the 'Effective Agglomeration Length' ($L_{eff}$), for a number of particle sizes. That equation appears on line 221. From the text one gathers that $\eta_{12}$ is proportional to the difference in the velocities of two particles, but the equation ignores the fact that those velocities have different phases in the oscillatory motion of the gas. A result that incorporates those phase differences was given by Temkin some time ago (Temkin, *Phys. Fluids*, 1994).

$$u_v - u_w = \frac{\omega\left( \tau_v - \tau_w \right)}{\sqrt{1 + \left( \omega \tau_v \right)^2} \sqrt{1 + \left( \omega \tau_w \right)^2}} \sin\left( wt' - \phi_v - \phi_w \right)$$

These velocities are scaled with the maximum air velocity in the wave (the same $U_0$ that appears without definition in the author's Eq. 1). Comparing this with the equation given by the authors we

see that $\eta_{12}$ refers to the *magnitude* of the velocity (or displacement) difference for two small spheres in a monochromatic sound wave. In any case, using absolute values is fine in a parametric study.

The issue here is that the authors parametrize the effect using particle size ranges and number concentrations that may or may not apply to clouds. It should be remembered that while a given size distribution specifies a size range, the opposite is not true. Particle size distributions in actual clouds are of considerable importance in the assessment of agglomeration effects produced in actual clouds by any effect. Incidentally, the theoretical and numerical work by Temkin cited above uses the coagulation equations to evaluate the agglomeration effects produced by a monochromatic sound wave on a specified particle-size distribution.

Returning to the article in review. One more parameter is needed before Eq. 1 can be used, and that is the gas velocity which the authors now call $U_g$ in the equation they give (without explanation) on line 225, which I re-write as follows:

$$\rho_g c * U_g = 10^{\left[(SPL - 94)/20\right]}$$

I am sure that at least some of your readers will, like myself, be mystified by this equation. So please forgive me for including here elementary material that de-mystifies it and that also tells us what is the meaning of its rhs, and what are the limitations imposed on the values of $U_g$ found this way.

To start, I assume that $c*$ is the sound speed in the gas in ambient conditions. If so, the product on the lhs has the dimensions of pressure, which in turns shows that the quantity on the rhs must be a pressure, or more specifically an acoustic pressure because the SPL appears there. Let's call that pressure $p'$. Its value follows from the definition of the SPL, namely, $SPL = 20\log_{10}\left(p'/P_{ref}\right)$, where $P_{ref}$ is the reference pressure, whose value is $2\text{x}10^{-5}$ N m$^{-2}$. Solving for $p'$ and using the fact that $20\log_{10}\left(P_{ref}\right) = 94$, we find that the rhs is equal to $p'$ so that the above equation can be written as

$$p' = \rho_g c * U_g$$

This is, of course, a limited form of a general theoretical result from acoustics that specifies the pressure fluctuation in a plane (unidirectional) acoustic wave, moving adiabatically with speed c* and inducing the fluid (a gas in this case) to move with speed $U_g$. The limitation here arises because $p'$ was derived from a SPL so that both $p'$ and $U_g$ are time-independent. In any case, the derivation shows that the gas velocity obtained from value of the SPL is not arbitrary but is constrained by whatever conditions apply to the equation above.

In addition, it should also be remembered that the equation shown on line 221, which uses that velocity, has its own set of requirements. That is, in addition to be limited by plane, isentropic sound waves, that equation is limited to waves in which the time variations of pressure and air velocity are sinusoidal. So far as I know the sounds generally associated with thunder do not fit that category.

In this context, the authors use some reported measurements of SPLs that are associated with the occurrence of 'thunder'. I have read some of the papers the authors mention and it is evident that there is no consensus as to what the measurements mean. Nor do we know whether they refer to a single, direct wave or to reflections or to some other effect. That is, a microphone will pick up whatever air vibrations exist in its immediate vicinity and will interpret them as an acoustic pressure. This means that the SPLs it reports cannot generally be tied to any specific motion. In other words, an SPL value cannot by itself tell us what were the air motions that produced it.

Sec. 3 Conclusions

Among the conclusions one finds the following statement (lines 14 and 15): "The results presented here demonstrate that thunder has the potential to alter the size distribution of cloud droplets in thunderclouds." For the reasons stated here this statement is not supported by the estimates of the effective agglomeration lengths presented in the article.

At the end of the article the authors bring up charge separation, obviously the most important issue as far as lightning is concerned. Their short discussion is hypothetical and I am not sure it belongs in the article. But charge separation due to droplet breakup is relevant and was considered some years ago by Dreyfuss and Temkin (Charge separation during rupture of small water droplets in transient flows: Shock tube measurements and applications to lightning, *J. Geophys. Res*. 88, C15. 1983

---

## Author Comment (AC1)

**Authors' response to reviewer #1**

Reviewer comments are in ***bold italics***.

"***In a note to the authors for future work, it may be worth estimating sound pressure levels from US NLDN data of peak lightning currents. Anecdotally, there are also indirect measurements from electric field changes of some negative lightning discharges, for example in Africa, with mega amp currents (personal communication with Phil Krider). These would translate to sound pressure levels of near 200 dB (energy E=I²RΔt assuming R=2 Ω and Δt = 50 μs, peak overpressure P=sqrt(2ρc²·Eacoustic) assuming Eacoustic is 0.01E, Lp =20·log(p/p₀))***": We thank the reviewer for the suggestion. Indeed, we envisage multiple aspects related to lightning and thunder that are worth investigating to refine parts of the calculations we present, including also the estimation of SPL level distribution and their occurrence frequency distribution.

"***Line 99–100: It's unclear where the We values stated derive from for droplets, ice crystals, etc.***": We rephrased "*For this particle size, even when the front velocity drops to 1 km s⁻¹, for a droplet We=167, for an ice particle We=63, and for a solid Al₂O₃ particle We=71*" to "*Substituting in the Weber number equation for this particle size, of 10 μm, and the corresponding surface tension (or surface energy), even when the front velocity drops to 1 km s⁻¹, for a droplet We=167, for an ice particle We=63, and for a solid Al₂O₃ particle We=71*".

"***Lines 123 and 292: It's unclear why Al₂O₃ and ice crystals must have 3 times the diameter of a water droplet to break up, if this could be explained further in this sentence***": It has to do with the surface tension (surface energy, for solids) $\sigma$, which in the Weber number equation $We=\rho g u 2 d/\sigma$ is in the denominator. $\sigma$ is higher in $Al_2O_3$ and ice crystals than in liquid droplets, so a higher $\sigma$ results in lower *We*. Setting *We* equal to 12 (critical Weber number) and solving for *d* (particle diameter), one gets $d=12\sigma\rho g u/2$, so for higher $\sigma$, *d* will also be higher. As $\sigma$ for water droplets is 0.072 N m⁻¹ while for water ice crystals it is 0.19 N m⁻¹ (2.64 times higher) and $Al_2O_3$ it is 0.169 N m⁻¹ (2.35 times higher), *d* will be correspondingly higher for the latter two. We now added in the sentence "This is not surprising, as setting *We* equal to 12 (critical Weber number) and solving the Weber equation for *d* (particle diameter), one gets $d=12\sigma\rho g u/2$, so for higher $\sigma$, *d* will also be higher. As $\sigma$ for water droplets is 0.072 N m⁻¹ while for water ice crystals it is 0.19 N m⁻¹ (2.64 times higher) and, e.g., $Al_2O_3$ it is 0.169 N m⁻¹ (2.35 times higher), *d* is correspondingly higher for the latter two".

"***Few spelling mistakes where "extent" is misspelled "extend": lines 134, 276, 298***": Spelling mistakes corrected.

---

## Author Comment (AC2)

**Authors' response to reviewer #2**

Reviewer comments are in ***bold italics***.

We were pleased but somewhat surprised to read this review. We were pleased for the thorough review and some constructive comments. We were surprised that the author of the review bypassed important parts of the conceptual framework of our manuscript and was largely focused on technical aspects of accuracy of approaches in acoustics. As a general remark on the general remarks of the reviewer, we note that physics, besides being an exact science, is also the art of approximation.

From the 5 works the reviewer cites, 3 are from Temkin, with which the reviewer seems very knowledgeable. Although we appreciate greatly the work done by Temkin, and we do cite some of his work in the manuscript, there is a variety of approaches in acoustics which is not exhausted by Temkin. We also do not exhaust these approaches in our manuscript, as our focus is different. As interesting a discussion about the foundations and theoretical approaches in fluid dynamics might be, we rather opt to have such a discussion elsewhere, as our manuscript does not aim at elucidating these.

Below our detailed response to the reviewer comments.

"***Main points (to be explained later) Contrary to the article's title and conclusions, the article does not consider, by far, the impact of thunder on cloud particles. At best it is a parametric study of two of the effects they consider:***": We will not go into linguistic arguments on the interpretation of language. We do insist that our paper considers effects of thunder on cloud particles, as the title suggests. We bring to the attention of the reviewer that the paper is not titled "**THE** impact of thunder …..", but "**ON THE** impact of thunder….", which would be a proper title even if the paper was "*at best […] a parametric study of two of the effects they consider*", as the reviewer finds. "***Breakup in shock waves and agglomeration in an acoustic wave. But there are problems with both. The first is considered on the basis of an erroneous assessment of the air velocity in a shock; the second….***": As these arguments are elaborated later on in the review, we will consider them below by replying to the elaborated reviewer arguments point after point.

"***Sec. 2.1*** …..***Table 1, which shows their estimates for the minimum sizes of particles that will break up, using as a criterion the experimentally-obtained value for the Weber number that produces droplet breakup, , which they say is 12. This value is in the range of values that have been found (although the actual value depends on the type of breakup). The problem I have in this regard is that they take that value to apply to the breakup of solid particles. I know of no experimental research supporting that application***": We=12 is not in the range of values that have been found, it is in the lower end of the range. We do not say arbitrarily that at Weber number 12 we will have droplet breakup. There is a large body of literature that supports this (e.g. Wierzba, 1990; Duan et al., 2003; Jain et., 2015; Strotos et al., 2016, to name a few). As to the type of breakup, we are not concerned with either the breakup type (bag breakup, bag-stamen breakup, multi-bag breakup or shear stripping breakup) or with the number of fragments that will be produced, but as we present

calculations for both We=12 and We=120, we do cover within this range all the abovementioned breakup types.

The fact that there are currently no experimental results on Weber number application to solid particles in gas flows, might have to do with hitherto limited possible applications for such studies. We are glad that our research might raise interest on the matter and prompt other researchers to study also experimentally the phenomenon in the lab and in the field.

Despite the absence of experimental results on Weber number for solid particles, it is a fact that there are no parameters in the Weber number equation that are only applicable in liquids and not to solids. The surface tension of a liquid parameterizes the attractive forces at the surface of the liquid, while the surface energy does exactly that for the attractive forces at the surface of a solid. When the kinetic energy on impact on a liquid particle is higher than the surface tension, we have droplet breakup. By analogy, for solid particles, when the kinetic energy on impact exceeds the surface energy, we see no physical reason for not having a breakup. After all, the surface energy of solids and the surface tension of liquids both relate to the energy state of the surface molecules and can be both expressed in N/m. Also, we note that ice crystals in cloud may or may not be entirely solid, i.e. snow crystals may also be present, for which the surface energy would be 0.03-0.72, depending on the liquid water content (e.g. Ketcham and Hobbs, 1969; Heil et al., 2020). We have now noted in the manuscript some of the above considerations ("*We note here, on the application of the critical Weber number to solid particles the following: Although there are (to our knowledge) no experimental determinations of the critical Weber number to solids, the surface energy of solids and the surface tension of liquids both relate to the energy state of the surface molecules and can be both expressed in N/m, hence it is possible to apply the same equation to solid particles. The modified Weber number We\*=We/12, equals the ratio of the kinetic energy on impact to the surface tension (surface energy, for solid particles). Hence, when the Weber number exceeds $We_{cr}$ , the kinetic energy on impact is higher than the surface tension (or energy), resulting in breakup. Also, we note that ice crystals in cloud may or may not be entirely solid, i.e. snow crystals may also be present, for which the surface energy would be 0.03-0.72, depending on the liquid water content (e.g. Ketcham and Hobbs, 1969; Heil et al., 2020). Additionally, $SiO_2$-methanol particles are not entirely solid*").

"***…they use the definition of the Weber number for breakup …. Here u is the air velocity. This is fine, but in getting the values of u corresponding to those two shock waves, the authors make a fundamental error: According to them (line 80) 'the relative air velocity equals the front velocity.' This is incorrect. The front of a shockwave always moves faster than the gas behind it. This is particularly important for the shock speeds they have in mind. The issue here is not the numerical difference but the erroneous conception of what a shock wave is***": We are well aware of what a shock wave is. It is a disturbance, but it is not a non-material disturbance. It is a disturbance of the medium (air, in our case) manifested through a very sudden change in pressure and density of air on its passage. Although the air behind the shock will be travelling indeed at very lower speeds than the shock front as the reviewer notes, as the shock front will result in an air pressure/density increase that will be travelling at the same speed as the front (because this is what the front is) and this shock disturbance will be sweeping through the cloud particles at that speed, there is no reason to

assume that the effect will be different than the effect of air travelling at the same speed: The air molecules at the shock wave front will be rapidly compressed (although they will decompress very quickly afterwards), and for this compression to move at the shock front speed, at the bow shock (the surface where the shock wave meets uncompressed air) the air molecules will have to move (even very briefly) at almost the same speed, at least for the nm distances that we are concerned here, i.e. the dimensions of the cloud particles. At 60 km/s, even if the air at the front will move at the front speed for just 1 ns, it will do so for a distance of 60 µm, which is much larger that most of cloud particles we are concerned here.

Additionally, as we already noted in the manuscript, "*During a lightning discharge, deposition of energy in the 4-100 J/cm range (Stark et al., 1996; Borovsky, 1998; Lacroix et al., 2019), heats within a few µs air to $10^4$-$10^5$ K plasma, resulting in very rapid expansion of air*". Hence, in the discharge column, air will expand very rapidly, and this is what will cause the shock wave. This very rapid expansion will move air very rapidly (at least as rapidly as the resulting shock front speed), hence at least in this area, we may even underestimate the impact on particles. In the discharge column the expansion of air within a few µs will be detonation-like. There is a lot of experimental evidence regarding shuttering even of whole man-made structures by detonations, so we see no reason to believe that ice or solid particles will be spared.

"***As for the shock speeds they use, 60 km/s and 1 km/s, I assume each may occur within a short distance from the lightning discharge,":*** Indeed, the assumption is correct. Shock speeds mentioned will occur within a short distance from the discharge. This is already discussed in the manuscript (lines 134-144 of the original manuscript), in as much detail as is now possible, given the present state of knowledge ("*With the extreme scarcity of data on the possible extend of the shockwave, it is not possible to evaluate how large are the parts of the cloud that are affected from the shockwave. Goyer and Plooster (1968) using a numerical model of lightning discharge, calculated shock waves in the order of a few meters. Karch et al. (2018) simulated a 96.4 kA strike (i.e. 0.76 X $10^4$ J $m^{-1}$) and found the shock wave transitioning to acoustic velocities at around 6 cm. Takagi et al. (1998) observed return lightning strokes with a high-speed camera and found that the luminous region expands at about 100 km $s^{-1}$ during the initial stage and reaches a maximum diameter of several meters after about 100 µs. If the Karch et al. (2018) 6 cm shockwave radius is used, then assuming a cylindrical geometry it is easy to calculate that the shockwave from a 500 m long intra-cloud (IC) discharge will affect a volume of 5.65 $m^3$ within the cloud. Although this volume is small, multiple IC lightning discharges are common within thunderclouds and will increase it substantially. If we use the Goyer and Plooster (1968) calculations, or if the shockwave extends at the same distance as the luminous region, we can assume that the shockwave extends ~3 m from the channel*"). ***"but I wonder about the effects on nm size ice particles of the high temperatures that exist behind the corresponding shocks,":*** we can only speculate here, for the sake of the present discussion, that in the very high temperatures (see Boggs et al., 2021, and references therein) of the lightning channel nm-sized ice particles will probably sublimate. Although this certainly may be a very interesting phenomenon to study, it is outside the scope of the present manuscript. In fact, many other phenomena may be at work within the pressure drop regions in the expanding chock wave, such as flash evaporation, which may even influence the shock wave itself (Mansur and Mueller, 2019), or

ice nucleation (Marcolli, 2017) but all these are matters with which we are not concerned here. "*as well as on the molecular composition of air in those conditions*": For one thing, production of NOx by lightning has been studied extensively, both experimentally and theoretically for some decades now. Changes in the molecular/ionic composition of air by lightning are still investigated, as the physics and chemistry of lightning are interesting for many disciplines and production of various gaseous species is possible (e.g. Barth et al., 2024, Karabulut, 2022). However, although the effects of the indeed very high temperatures of the lightning channel on the molecular composition of air are very interesting, they are outside the scope of our paper and we do not understand why the reviewer brings this up.

"*To conclude, there is nothing new in this section other than an erroneous estimate of the minimum droplet sizes that result in breakup and the questionable use of information obtained with water droplets for the case of solid particles*": On the "*questionable use*" we have replied above. "*erroneous estimate*" and "*there is nothing new here*" are quite arbitrary statements. There have been no other estimates than the ones presented in our manuscript on the possible impact of this mechanism in the real atmosphere.

"*Sec. 2.2 Particle agglomeration in the thunder sonic field*"

"*In this section the issue of agglomeration in 'the thunder sonic field' is parametrized in the basis of a procedure, called orthokinetic agglomeration by some authors*": To our knowledge, it is called so by everybody, not just "*some authors*". "*Although there are several variations, the technique refers to agglomeration of particles due to the relative velocities of small particles of different sizes responding to single-frequency sound wave. The literature is vast but a basic source is Mednikov, Acoustic coagulation and precipitation of Aerosols, Consultant Bureau, 1965*": Although we already cited relevant work, we added now also the proposed reference. "*Most of the work in this area has been done with high-frequency sound waves*": Yes, most of the work in this area has been done with high frequency waves, but substantial work has also been done with low frequency waves, e.g.: Hoffmann and Koopmann, 1996 (400-900 Hz), Liu et al., 2009 and Cheng et al., 1983 (600 Hz), while Dong et al. (2006) have also addressed frequencies well below 100 Hz. Although orthokinetic agglomeration might be most effective at the 100-200 Hz range, it is also quite effective also below 100 Hz (e.g. Dong et al., 2006). We do address this ranges in our manuscript. "*but some time ago Temkin (Droplet agglomeration in a shock wave flow field, Phys. Fluids, June1570, not cited) showed that agglomeration can be effectively produced with low frequency waves that have shock fronts*": We do not understand why the reviewer brings this up in this section of the review, which deals with Sec. 2.2 of our manuscript, as shock waves are treated in the previous section of our manuscript (Sec. 2.1). The reviewer gives wrong date for the publication (1570 instead of 1970), but also the mentioned title is incorrect, as the title of the Temkin manuscript is not "*Droplet agglomeration in a shock wave flow field*" but actually it is "Droplet Agglomeration Induced by Weak Shock Waves". Anyway, in section 2.1 we are not concerned with weak shock waves causing agglomeration (as in the publication mentioned by the reviewer) but with strong ones causing shattering, and in section 2.2 we are not examining this mechanism, as we are also not examining other possible mechanisms for acoustic agglomeration, such as mutual scattering interaction, mutual radiation pressure interaction and acoustic wake

influence. We examine only the orthokinetic agglomeration mechanism. We do not claim in the manuscript to exhaust the matter on agglomeration mechanisms in thunderclouds. In this regard, we find the mentioned publication not very relevant to this section of our manuscript.

"*As remarked later, much closer to the work in review is Temkin's study of agglomeration of polydisperse distributions by monochromatic sound waves (Temkin, Gasdynamic agglomeration of aerosols. I, Acoustic waves, Phys. Fluids, July 1994, not cited)*": Although we cited already 3 works by Temkin in the manuscript, the ones we considered most relevant to our work, we now added this one also.

"*In this work, the authors use the term 'resonance ratio' for the ratio of the displacement (or velocity) of a particle in a sound field to the gas velocity in the wave, usually called the 'Entrainment Ratio' in the literature":* We use the term resonance rate, not ratio. This is referred also as entrainment coefficient (not "*entrainment ratio*") in the literature (e.g. Gonzalez et al., 2000; Shen et al., 2018). We now included both terms in the manuscript. *"In any event, the ratio as given by eq. 1 […] Note that the two velocities appearing in this equation are not defined,":* This was an inadvertent omission. Definition of the two velocities are now included in the manuscript ("*where Up is the particle velocity amplitude and Uo the gas velocity amplitude*"). "*nor is it stated anywhere in the text that the equation applies only in very limited cases: spherical particles, single-frequency sound waves, very small gas velocities (small Reynolds number) and frequencies that are not large*": Cloud droplets ARE spherical particles. Cloud ice crystals may or may not be spherical. While in certain cloud regions ice crystal shapes may be dominated by spheroids (e.g. Lawson et al., 2001; Fleishauer et al., 2002), a variety of other shapes will dominate in other regions. As an approximation, we believe we can apply the mentioned equation, as currently there is nothing better at hand. We also applied the equation for single frequencies. We did not apply the equation to very high frequencies. Regarding the gas velocities, at parts of the cloud the Reynolds number will be low and at parts, it will be high, i.e. the flow will be turbulent. Regarding turbulence's influence on agglomeration in the presence of sound, this may be the subject of future work elaborating on the matter, but as an exploratory attempt our approach may serve as a first approximation. After all, effects of turbulence are still under investigation even for cloud processes that are studied for decades (e.g. Shawon et al., 2021). Also, we note that possible effects of turbulence on the calculations, may also depend on the mean flow velocity (Zhao et al., 2019), which is not large in our case. In any way, we now included in the manuscript a note on the inherent assumptions for eq. 1 ("*We note here that the validity of eq. (1) has been demonstrated for spherical particles, single-frequency sound waves, small gas velocities and relatively low frequencies and the above assumptions may not hold throughout the cloud region. However, it can still give us an idea of the involved processes, as we apply here the equation to relatively low single frequencies, and cloud droplets are spherical, while cloud ice crystals may or may not be spherical. While in certain cloud regions ice crystal shapes may be dominated by spheroids (e.g. Lawson et al., 2001; Fleishauer et al., 2002), a variety of other shapes will dominate in other regions*"). "*In any event, the issue is not the way τp is written or why is it called that way, but whether Eq. 1 is at least approximately valid for a parametric study of effects that supposedly take place in a cloud as a result of thunder*": This we will leave to following experimental

laboratory and field studies to show, as, at this stage, one can only speculate on the validity not only of the present approximation but also for the validity of other approximations, as the one by Temkin (1994) the reviewer cites in the beginning of this paragraph of the review. Also, we do not claim to settle the matter, we only demonstrate in our manuscript some basic features the involved processes might have.

"***In any case, Eq. 1 is apparently used in the article to obtain the basic relation that the authors use to estimate the 'Effective Agglomeration Length' (Leff), for a number of particle sizes. That equation appears on line 221. From the text one gathers that η₁₂ is proportional to the difference in the velocities of two particles, but the equation ignores the fact that those velocities have different phases in the oscillatory motion of the gas. A result that incorporates those phase differences was given by Temkin some time ago (Temkin, Phys. Fluids, 1994)….***": The reviewer here refers to "S. Temkin; Gasdynamic agglomeration of aerosols. I. Acoustic waves. *Physics of Fluids* 1 July 1994; 6 (7): 2294–2303. https://doi.org/10.1063/1.868180", which we now include in the references. Very different approaches exist in the field of acoustics as to possible refinement of results. Whether including phase differences, as Temkin did, will indeed result in much different or more accurate results is a matter of question. The Temkin 1994 work, has its own limitations, as we are sure the reviewer also knows. As stated by Temkin in the work mentioned by the reviewer, it uses "*the simplest possible conditions*", ignores turbulence, uses "*plane, monochromatic sound wave*", ignores particles below 350 nm and above 2 μm, "*is only meant to illustrate the main features of the process*", and "equally important is the need to understand acoustic agglomeration in the simplest possible conditions, without any of the effects mentioned above" (the latter referring to turbulence, steady streaming, drift resulting from nonlinear behavior, particle interactions). After all, we also intend to show some main features of the involved processes in simple conditions, so we think the reviewer does not need to be so critical about possible assumptions and limitations in our work, since at the same time the reviewer is proposing us approaches by Temkin that have their own assumptions and limitations.

"***The issue here is that the authors parametrize the effect using particle size ranges and number concentrations that may or may not apply to clouds. It should be remembered that while a given size distribution specifies a size range, the opposite is not true. Particle size distributions in actual clouds are of considerable importance in the assessment of agglomeration effects produced in actual clouds by any effect. Incidentally, the theoretical and numerical work by Temkin cited above uses the coagulation equations to evaluate the agglomeration effects produced by a monochromatic sound wave on a specified particle-size distribution***": As said above, we now include the proposed work by Temkin in the references. We note that the cloud droplet size ranges we used (8-36 μm) are well applicable in clouds (see references in the manuscript), whereas the ones used in the Temkin work (0.5-1.5 μm) may not be. We agree with the reviewer that actual size distributions are of importance, but we intent here to show just some main features of the process and we do not claim to produce very accurate results.

"***…, before Eq. 1 can be used, and that is the gas velocity which the authors now call in the equation they give (without explanation) on line 225, which I re-write as follows [...] I am***

*sure that at least some of your readers will, like myself, be mystified by this equation. So please forgive me for including here elementary material that de-mystifies it and that also tells us what is the meaning of its rhs, and what are the limitations imposed on the values of found this way. To start, I assume c\* that is the sound speed in the gas in ambient conditions*": The assumption is correct, but this was already stated in line 226 of the manuscript, "c the velocity of sound in air" (not c\*, \* was the multiplication sign in the equation).

"*This is, of course, a limited form of a general theoretical result from acoustics that specifies the pressure fluctuation in a plane (unidirectional) acoustic wave, moving adiabatically with speed c\* and inducing the fluid (a gas in this case) to move with speed Ug. The limitation here arises because p' was derived from a SPL so that both p' and Ug are time-independent. In any case, the derivation shows that the gas velocity obtained from value of the SPL is not arbitrary but is constrained by whatever conditions apply to the equation above. In addition, it should also be remembered that the equation shown on line 221, which uses that velocity, has its own set of requirements. That is, in addition to be limited by plane, isentropic sound waves, that equation is limited to waves in which the time variations of pressure and air velocity are sinusoidal. So far as I know the sounds generally associated with thunder do not fit that category*":

"*In this context, the authors use some reported measurements of SPLs that are associated with the occurrence of 'thunder'. I have read some of the papers the authors mention and it is evident that there is no consensus as to what the measurements mean*": We are not sure what the reviewer means by "what the measurements mean". Obviously, the comment is not on what the SPL measurements mean, as there is tremendous consensus as to what SPL means. If the reviewer here refers to the following comment *"Nor do we know whether they refer to a single, direct wave or to reflections or to some other effect"*, we again do not understand what the difference would be if it was a direct wave or a reflected one. It will in cases be a direct wave with some contribution from reflections, the extend of the reflections contribution depending on the topography and lightning channel geometry. *"That is, a microphone will pick up whatever air vibrations exist in its immediate vicinity and will interpret them as an acoustic pressure. This means that the SPLs it reports cannot generally be tied to any specific motion. In other words, an SPL value cannot by itself tell us what were the air motions that produced it*": So what? We may infer here that the reviewer with "*there is no consensus as to what the measurements mean*", may be concerned with the nature of the air oscillatory motions producing the SPL. Let's name here Fpeak the peak audio frequency of a stroke. Fpeak, in most of the works we cited, has a power that may be 5 times larger than the power of frequencies Fpeak/2 and 2XFpeak, so it is not unreasonable to note that there is a very large power of the disturbance that is due to a sinusoidal sound wave at Fpeak.

"*Sec. 3 Conclusions*"

"*"The results presented here demonstrate that thunder has the potential to alter the size distribution of cloud droplets in thunderclouds." For the reasons stated here this statement is not supported by the estimates of the effective agglomeration lengths presented in the*

*article*": For the reasons we state above we insist that thunder has the potential to alter the size distribution of cloud droplets.

"***At the end of the article the authors bring up charge separation, obviously the most important issue as far as lightning is concerned. Their short discussion is hypothetical and I am not sure it belongs in the article. But charge separation due to droplet breakup is relevant and was considered some years ago by Dreyfuss and Temkin (Charge separation during rupture of small water droplets in transient flows: Shock tube measurements and applications to lightning, J. Geophys. Res. 88, C15, 1983***": The reviewer first names our short discussion on charge separation "*hypothetical*", although he finds it "*most important*" and "*relevant*" and cites some work on the matter (also by Temkin, by the way). What we say is "*As the charge separation within thunderclouds is influenced by the size distribution of cloud particles and the collisions between them, it is interesting that this charge separation, causing lightning, also causes thunder that results then in collision enhancement and changes in the size distribution and hence might introduce a yet unquantified feedback in the electrification process*". The work of Dreyfuss and Temkin deals with charge separation by the breakup of 1-2 mm droplets behind a shock wave, and as such may be of some relevance to the discussion here, although in this size range there are no cloud droplets but rather raindrops. Iribarne and Klemes (1970) and Zilch et al. (2008) might be more relevant, as they studied µm sized droplets. Anyway, it is well known that droplet breakup produces charge separation (e.g. Jonas and Mason, 1968), and we do not want to extend the discussion on this, we just bring it up briefly to the attention of the atmospheric community, so as to simulate some future research.

**Final remarks by the authors on the review**

We agree with the reviewer that there is a need for more future work to be done on various aspects (the reviewer notes many of those) of the calculations we present here and we are hoping that the present work will simulate these.

**References**

Boggs, L. D., Liu, N., Nag, A., Walker, T.D., Christian, H. J., da Silva, C. L., et al. (2021). Vertical temperature profile of natural lightning return strokes derived from optical spectra. Journal of Geophysical Research: Atmospheres, 126, e2020JD034438. https://doi.org/10.1029/2020JD034438

Barth P., Stüeken E., Helling C., Schwieterman E.W., Telling J., The effect of lightning on the atmospheric chemistry of exoplanets and potential biosignatures, Astronomy and Astrophysics, 686, A58, 2024, https://doi.org/10.1051/0004-6361/202347286

Cheng M.T., P.S Lee, A Berner, D.T Shaw, Orthokinetic agglomeration in an intense acoustic field, Journal of Colloid and Interface Science, 91:1, 176-187, 1983, https://doi.org/10.1016/0021-9797(83)90324-7

Dong S., B. Lipkens, T.M. Cameron, The effects of orthokinetic collision, acoustic wake, and gravity on acoustic agglomeration of polydisperse aerosols, Journal of Aerosol Science, Volume 37, Issue 4, 2006, Pages 540-553, https://doi.org/10.1016/j.jaerosci.2005.05.008.

Duan R.-Q., S. Koshizuka, Y. Oka, Numerical and Theoretical Investigation of Effect of Density Ratio on the Critical Weber Number of Droplet Breakup, Journal of Nuclear Science and Technology, 40:7, 501-508, DOI:10.1080/18811248.2003.9715384, 2003.

Fleishauer R.P., Larson V.E., Haar T.H.V. Observed microphysical structure of midlevel, mixed-phase clouds, J. Atmos. Sci., 59 (11) (2002), pp. 1.779-1.804

González I., Thomas L Hoffmann, Juan A Gallego, PRECISE MEASUREMENTS OF PARTICLE ENTRAINMENT IN A STANDING-WAVE ACOUSTIC FIELD BETWEEN 20 AND 3500 Hz, Journal of Aerosol Science, 31(12), 1461-1468, 2000, https://doi.org/10.1016/S0021-8502(00)00046-X.

Heil, J.; Mohammadian, B.; Sarayloo, M.; Bruns, K.; Sojoudi, H. Relationships between Surface Properties and Snow Adhesion and Its Shedding Mechanisms. *Appl. Sci.* **2020**, *10*, 5407. https://doi.org/10.3390/app10165407

Hoffmann, T.L., G.H. Koopmann, Visualization of acoustic particle interaction and agglomeration: Theory and experiments, J. Acoust. Soc. Am., 99, 2130, https://doi.org/10.1121/1.415400, 1996.

Iribarne, J. V., and M. Klemes, Electrification Associated with Breakup of Drops at Terminal Velocity in Air. J. Atmos. Sci., 27, 927–936, https://doi.org/10.1175/1520-0469(1970)027<0927:EAWBOD>2.0.CO;2 , 1970.

Jain, M., Prakash, R.S., Tomar, G., and Ravikrishna, R.V.: Secondary breakup of a drop at moderate Weber numbers, Proc. R. Soc. A, 471: 20140930, http://dx.doi.org/10.1098/rspa.2014.0930, 2015.

P. R. Jonas, B.J. Mason, Systematic charging of water droplets produced by break-up of liquid jets and filaments, Trans. Faraday Soc., 64, 1971-1982, 1968.

Karabulut E., Oxygen Molecule Formation and the Puzzle of Nitrogen Dioxide and Nitrogen Oxide during Lightning Flash, The Journal of Physical Chemistry A, 126 (32), 5363-5374, 2022, DOI: 10.1021/acs.jpca.2c02378

Ketcham, W.M.; Hobbs, P.V. An experimental determination of the surface energies of ice. Philos. Mag. J. Theor. Exp. Appl. Phys. **1969**, 19, 1161–1173.

Lawson R.P., Baker B.A., Schmitt C.G., Jensen T.L. An overview of microphysical properties of Arctic clouds observed in May and July 1998 during FIRE ACE J. Geophys. Res-Atmos., 106 (D14) (2001), pp. 14.989-15.014

Liu J., G. Zhang, J. Zhou, J. Wang, W. Zhao, K. Cen, Experimental study of acoustic agglomeration of coal-fired fly ash particles at low frequencies, Powder Technology, 193:1, 20-25, 2009, https://doi.org/10.1016/j.powtec.2009.02.002.

Mansour A., N. Müller, A review of flash evaporation phenomena and resulting shock waves, Experimental Thermal and Fluid Science, 107, 146-168, 2019, https://doi.org/10.1016/j.expthermflusci.2019.05.021.

Marcolli, C. Ice nucleation triggered by negative pressure. *Sci Rep* **7**, 16634 (2017). https://doi.org/10.1038/s41598-017-16787-3

Shawon, A. S. M., Prabhakaran, P., Kinney, G., Shaw, R. A., & Cantrell, W. (2021). Dependence of aerosoldroplet partitioning on turbulence in a laboratory cloud. *Journal of Geophysical Research: Atmospheres*, *126*, e2020JD033799. https://doi.org/10.1029/2020JD033799

Shen G., Xiaoyu Huang, Chunlong He, Shiping Zhang, Liansuo An, Experimental study of acoustic agglomeration and fragmentation on coal-fired ash with different particle size distribution, Powder Technology, 325, 145-150, 2018, https://doi.org/10.1016/j.powtec.2017.10.037.

Strotos G., I. Malgarinos, N. Nikolopoulos, M. Gavaises, Predicting droplet deformation and breakup for moderate Weber numbers. International Journal of Multiphase Flow, 85, 96-109, 2016. doi: 10.1016/j.ijmultiphaseflow.2016.06.001

Wierzba A., Deformation and breakup of liquid drops in a gas stream at nearly critical Weber numbers. Experiments in Fluids 9, 59–64 (1990). https://doi.org/10.1007/BF00575336

Zhao H., D. Nguyen, D. J. Duke, D. Edgington-Mitchell, J. Soria, H.-F. Liu, D. Honnery, Effect of turbulence on drop breakup in counter air flow, International Journal of Multiphase Flow, 120, 103108, 2019, https://doi.org/10.1016/j.ijmultiphaseflow.2019.103108.

Zilch L.W., J.T. Maze, J.W. Smith, G.E. Ewing, M.F. Jarrold, Charge Separation in the Aerodynamic Breakup of Micrometer-Sized Water Droplets, *The Journal of Physical Chemistry A, 112* (51), 13352-13363, 2008, DOI: 10.1021/jp806995h